# Vaccinomics-Aided Development of a Next-Generation Chimeric Vaccine against an Emerging Threat: *Mycoplasma genitalium*

**DOI:** 10.3390/vaccines10101720

**Published:** 2022-10-14

**Authors:** Kashaf Khalid, Tajamul Hussain, Zubia Jamil, Khalid Salman Alrokayan, Bashir Ahmad, Yasir Waheed

**Affiliations:** 1Clinical and Biomedical Research Center, Foundation University Medical College, Foundation University Islamabad, Islamabad 44000, Pakistan; 2Research Chair for Biomedical Application of Nanomaterials, Biochemistry Department, College of Science, King Saud University, Riyadh 11451, Saudi Arabia; 3Center of Excellence in Biotechnology Research, College of Science, King Saud University, Riyadh 11451, Saudi Arabia; 4Department of Medicine, Foundation University Medical College, Foundation University Islamabad, Islamabad 44000, Pakistan; 5College of Medicine, Alfaisal University, Riyadh 11533, Saudi Arabia; 6Department of Biotechnology, International Islamic University, Islamabad 44000, Pakistan; 7Office of Research, Innovation and Commercialization, Shaheed Zulfiqar Ali Bhutto Medical University (SZABMU), Islamabad 44000, Pakistan; 8Gilbert and Rose-Marie Chagoury School of Medicine, Lebanese American University, Byblos 1401, Lebanon

**Keywords:** *Mycoplasma genitalium*, subunit vaccine, epitopes, T-lymphocytes, molecular docking, molecular dynamic simulations

## Abstract

*Mycoplasma genitalium*, besides urethritis, causes a number of other sexually transmitted diseases, posing a significant health threat to both men and women, particularly in developing countries. In light of the rapid appearance of multidrug-resistant strains, *M. genitalium* is regarded as an emerging threat and has been placed on the CDC’s “watch list”. Hence, a protective vaccine is essential for combating this pathogen. In this study, we utilized reverse vaccinology to develop a chimeric vaccine against *M. genitalium* by identifying vaccine targets from the reference proteome (Strain G-37) of this pathogen. A multiepitope vaccine was developed using proteins that are non-toxic, non-allergic, and non-homologous to human proteins. Several bioinformatic tools identified linear and non-linear B-cell epitopes, as well as MHC epitopes belonging to classes I and II, from the putative vaccine target proteins. The epitopes that showed promiscuity among the various servers were shortlisted and subsequently selected for further investigation based on an immunoinformatic analysis. Using GPGPG, AAY, and KK linkers, the shortlisted epitope sequences were assembled to create a chimeric construct. A GPI anchor protein immunomodulating adjuvant was adjoined to the vaccine construct’s N-terminus through the EAAK linker so as to improve the overall immunogenicity. For further investigations of the designed construct, various bioinformatic tools were employed to study the physicochemical properties, immune profile, solubility, and allergenicity profile. A tertiary chimeric design was computationally modeled using I-TASSER and Robetta and was subsequently refined through GalaxyRefine. ProSA-Web was exploited to corroborate the quality of the construct by detecting errors and the Ramachandran plot was used to identify possible quality issues. Simulation studies of the molecular dynamics demonstrated the robustness and flexibility of the designed construct. Following the successful docking of the designed model to the immune receptors, the construct was computationally cloned into Escherichia coli plasmids to affirm the efficient expression of the designed construct in a biological system.

## 1. Introduction

*Mycoplasma genitalium* has successfully established itself as a relatively new sexually transmitted infectious (STI) agent that has devastating health effects on men and women alike [1]. The increasing prevalence of multidrug-resistant *M. genitalium* has made it the most common cause of STI, second to *Chlamydia trachomatis* [2]. A CDC report has recently put this bacteria on the “watch list” of antibiotic resistance threats, due to its rapid development of resistance to many antibiotics, including macrolides and fluoroquinolones used for treating other STIs [3,4]. *M. genitalium* infections are conducive to urethritis, cervicitis, endometritis, pelvic inflammation, tubal factor infertility, preterm delivery, and ectopic pregnancy [5]. In general, 15% to 20% of cases of nongonococcal urethritis (NGU), 20% to 25% of non-chlamydial urethritis (NCU), and 40% of recurrent or persistent urethritis are caused by the bacterium [6].

*M. genitalium* is a slow-growing, Gram-negative, pleiomorphic, and obligate intracellular facultative anaerobe. With a genome of 580 kb, 32.2% GC content, and 485 protein-coding genes, the prokaryote is considered as the tiniest organism capable of self-replication [5,7]. Initially discovered from a male with urethritis by Tully et al. in 1981 [8], the entire genome of the bacterium was sequenced for the first time in 1995 [9].

As opposed to other Gram-negative bacteria, *M. genitalium* is devoid of cell walls, which makes it inherently resistant to the class of antibiotics that inhibit peptidoglycan assembly, such as penicillin and cephalosporins [10]. Reports suggest that *M. genitalium* can partake in pathogenicity as a sexually transmitted agent or as a co-infection facilitator [5]. In view of the bacterium’s increasing resistance, its treatment has proven to be onerous in clinical terms [11]. Despite the progress in medicine in general, *M. genitalium’s* resistance towards azithromycin and moxifloxacin, which have been recommended by the CDC for the treatment of STDs, has become commonplace [12,13,14]. As a consequence of the aforementioned reason, the bacterium has gained notoriety as an “emerging STI” by the CDC and others [15,16,17,18,19,20,21].

The failure of current therapeutic approaches in dealing with *M. genitalium* is attributed to a variety of reasons, including the absence of a global agreement regarding the treatment policy, the injudicious use of macrolides for community-acquired pneumonia, concomitance with HIV infection, and poor compliance to therapeutic regimens by both patients and their partners [10,22,23,24]. As a result of bacterial evolution and escape from the host’s immune surveillance, it is often suggested that antigenic variation is the most commonly employed strategy for combating bacteria [25]. Several reports suggest the vaccine approach to be a plausible solution in combating such STI agents [26]. A traditional vaccine is designed with large proteins, and the incorporation of inappropriate antigens is likely to lead to hypersensitivity; however, a multiepitope-based vaccine employing shorter peptide fragments and eliciting an efficient immune response might overcome these limitations [27]. The expedient and cost-effective nature of this method reduces the duration of the overall therapeutic development process [28]. Currently, none of the Federal Drug Administration’s approved *M. genitalium* vaccines are available to humans [6].

The complexity of bacterial antigens and their ability to evade host immunity make multiepitope vaccines more likely to trigger a strong and broader immune response. In the current study, an immunoinformatic stratagem was implemented in a progressive fashion to propose a multiepitope subunit vaccine by identifying immune-potentiating, non-allergic, and non-toxic regions of the bacterium. Using linkers, potential epitopes were connected to an effective adjuvant to create a chimeric vaccine. The final chimeric construct contained B-cell epitopes (BCEs) and cytotoxic T and helper T lymphocytes (CTL and HTLs). Furthermore, a physicochemical analysis, docking studies, and a stability analysis of the construct were conducted to gauge its safety and effectiveness. The vaccine candidates selected through computational analysis could be tested in wet labs, which may aid in diminishing *M. genitalium*-induced infections.

## 2. Materials and Methods

An outline of the process employed for developing an effective vaccine candidate is depicted in Figure 1.

### 2.1. Retrieval of M. genitalium Reference Proteome

The reference proteome of *M. genitalium* (G-37/strain ATCC 335300/NCTC 10195) was retrieved from UniProt [29]. The reference strain was selected because it is well conserved, more common, and clinically significant in comparison to other *M. genitalium* and Mycoplasmas [30]. A file containing the FASTA sequences of each of the proteins was then uploaded to the VaxiJen server that employs an alignment-independent algorithm based on primary amino acid characteristics [31]. The proteins were ranked and filtered on the basis of their antigenic scores (>0.5). Subsequently, the subcellular localization was used to shortlist the proteins using a web server (CELLO2GO) [32]. This filter shortlisted the proteins to a total of 7.

### 2.2. Epitope Mapping

The NetCTL webserver was used to screen for the most immunogenic HLA I epitope sequences among the seven shortlisted putative vaccine candidates against 12 HLA super-families [33]. The CTLs were predicted via this server using the preset parameters and thresholds (MHC super type A1: 0.75; TAP transport efficiency: 0.05; C-terminal cleavage: 0.15). We employed the IEDB prediction server to determine the HLA class II epitope binding, using percentile rank and IC50 values as the prioritization criteria [34]. The epitopes for B cells could be characterized into two groups: linear (continuous) and non-linear (or discontinuous). The ABCpred [35], BCpred [36], and Bepipred [37] servers identified linear BCEs, while a conformational BCE analysis was performed using the SvmTrip webserver [38]. Appendix A lists 280 selected class I and class II HLA epitopes and BCEs in each of the selected proteins.

### 2.3. Conserved Epitope Analysis

A BlastP analysis was conducted against proteomes to study the homologous sequences in humans as well as in other pathogenic species of Mycoplasma. Sequences showing 70% identity and 40% query coverage depicted the homologous sequences. In addition, a conserved epitopes analysis was performed using the IEDB conservation analysis tool. The tool computes the level of conservation of an epitope in a protein dataset. Using conservation as an indicator, we could compute the percentage of sequences in a protein that contained the same epitopes at a certain level of identity.

### 2.4. Epitopes Assembly

To develop the final vaccine, epitopes generated from multiple immunoinformatic programs were adjoined using GPGPG, AAY, and KK linkers, which play a significant role in providing flexibility, folding, stability, and functional domain separation [39]. According to a number of studies, *M. genitalium* infections stimulate cytokine production by human genital epithelial cells by triggering TLR-2 and TLR-6 [40,41]. The TLR 2/6 agonist, GPI anchor protein of *Trypanosoma cruzi* (Uniprot ID: P84883), was affixed as an adjuvant via the EAAK linker to the N-terminus of the designed chimera to enhance the overall immunogenic properties [42,43].

### 2.5. Profiling of Immunogenicity, Allergenicity, and Physicochemical Characteristics

Using two web servers—VaxiJen and ANTIGENpro—we analyzed the antigenicity of the designed construct with and without the adjuvant [31,44]. The designed construct was evaluated using AlgPred and AllerTop to determine whether it is an allergen or not. The protein was subsequently examined based on physicochemical considerations using the computational tool ProtParam [45]. A wide range of properties were studied, including the chemical formula, stability index, molecular weight, isoelectric point, half-life, and GRAVY value.

### 2.6. Solubility Analysis

In order to ensure that a protein is soluble when overexpressed in *E. coli*, it is crucial to determine the solubility of the chimeric construct. For this purpose, an online tool SOLpro was employed [46]. The server operates at a cut-off value of 0.5 and generates data with an accuracy of 74.15%. For further validation of the solubility, Protein-sol, a web-based suite, designed for computational prediction of scaled solubility value (QuerySol) with respect to the experimental dataset (PopAvrSol) [47]. The server uses a cut-off value of 0.45, where higher values demonstrate that the protein is more soluble as compared to the average soluble *E. coli* protein.

### 2.7. Computation of Secondary Structure

We forecasted the secondary structure of the designed construct using PSIPRED v3.3 [48]. The server employs a strict cross-validation method to accomplish the prediction of coils and alpha and beta helices, with an accuracy rate of 81.6%.

### 2.8. Computation of Tertiary Structure (3D)

The 3D modeling of the proposed construct was performed to generate a good-quality tertiary structure using two webservers, I-TASSER and Robetta [49,50]. The I-TASSER server is an organized platform, the working protocol of which incorporates four stages: (i) threading; (ii) structural assembly; (iii) model selection and refinement; (iv) structure-based functional annotation. The top five models are generated, out of which the model showing the highest confidence score is prioritized. The C-scores generally fall in a range of −5 to 2 [49]. Alternatively, Robetta parses the submitted sequence into presumed domains and constructs the structure using either de novo or comparative modeling methods [50].

### 2.9. 3D Model Validation and Refinement

With the predicted tertiary structures from I-TASSER and Robetta, we analyzed the Ramachandran plot values to select one structure for further refinement. The model showing more values in the core region was selected to refine the structural quality locally and globally [51]. The GalaxyRefine webserver employed dynamic simulations to carry out rigorous structural perturbations and relaxation [52]. To validate the refined structure, the percentages of residues in allowed, disallowed, and favored regions of the Ramachandran plot were visualized using the graph generated by the RAMPAGE webserver [53].

### 2.10. Probing of Discontinuous BCEs

The discontinuous BCEs (>90% compared to other epitopes), playing a crucial role in vaccine design, are formed as a result of the spatial configuration that brings together disparate residues [54]. In a bid to validate the presence of these essential epitopes, we used the Ellipro webserver [55]. The sever applies Thornton’s method to perform residue clustering and finally enables the visualization of the antibody epitopes via the Jmol view.

### 2.11. Docking Studies of the Modeled Vaccine with Toll-like Receptors

By using the molecular docking technique, we observed the preferred orientation of the ligand and the affinity of its binding to the human immune receptors [56]. The immune response prompted by *M. genitalium* primarily activates TLRs 1, 2, and 6 [57]. We, thus, selected TLR 1, TLR 2, and TLR 6 (PDB ids: 6NIH, 2Z7X, 4OM7) as docking receptors and the refined vaccine 3D model as the ligand. Our study employed the ClusPro 2.0 server [58], HDOCK [59], and PatchDock [60]; PatchDock yielded numerous docking solutions, which were then subjected to refinement using FireDock [61]. With the PRODIGY webserver, we further explored the binding affinity of the docked complexes [62]. This webserver not only predicted the binding affinities but also the interfaces in biological complexes. Finally, the online available tool PDBsum was harnessed to visually demonstrate the interactions between docked complexes [63].

### 2.12. Energy Optimization and Simulation of Molecular Dynamics

In addition to energy minimization, molecular dynamic (MD) studies were performed using a Linux-based environment GROMACS [64]. The MD study was carried out to understand the behavior of the designed construct in life-like environments. Based on the OPLS-AA force field constraint, the topology file required to minimize the energy and overall equilibrium was obtained. A simulation of the vaccine using periodic boundary conditions was performed using an equilibrated three-point water model, spc216. To neutralize the vaccine construct, the net charge was gauged to add further ions. Additionally, to calculate the RMSD and RMSF of the backbone and side chain, respectively, we simulated the energy-minimum structure for 50 ns. The Xmgrace plotting tool was exploited to visualize the graphs [65].

### 2.13. Simulated Immune Responses

C-ImmSimm was used for immune simulations of the final candidate vaccine to analyze immune responses [66]. The webserver operates by utilizing a position-specific scoring matrix (PSSM) to calculate the amount of immune responses generated. The candidate vaccine was simulated using all pre-set parameters with time steps at 1, 42, and 84 [67]. Thus, 1050 simulation steps were completed in total after three injections were given simultaneously.

### 2.14. Reverse Transcription and Computational Cloning into Vector Backbone

In order to construct an effective plasmid carrying the multiple epitope sequence, the JCat server was used [68]. An organism-specific and codon-adjusted form of the DNA of interest is provided on this server. Besides the code adaptation index (CAI), the output includes the GC content. Ideally, the CAI should be 1.0 and th GC content should fall between 30 and 70%. SnapGene software was harnessed to successfully carry out the cloning into the pET28a (+) expression vector. Furthermore, *NdeI* and *XhoI* restriction sites were incorporated to ensure the opposite addition of the designed chimera into the pET28(a)+ vector.

## 3. Results

### 3.1. Finalization of Protein Sequences for Development of Vaccine Based on Immunogenicity Analysis

The proteome of *M. genitalium* was retrieved, comprising 483 proteins. As a prerequisite for the development of a subunit vaccine, candidate proteins that promote a protective immune response must be identified [69]. Accordingly, the protein sequences were transmitted to the VaxiJen server for antigenicity determination using the antigenic scores. From the 483 proteins with antigenicity scores greater than 0.5, for 138 we further filtered the proteins according to subcellular location using CELLO2GO and PSORTb, which resulted in a list of 50 proteins (Appendix A). The top 50 proteins were analyzed further to assess the existence of additional criteria (TM α-helices, signal peptides, essentiality, and virulence) to capsulize the best seven proteins as the probable vaccine candidates. As a result, new antigenic proteins were found that have not yet been analyzed. These consist of SecG (protein export membrane protein), PLSY (glycerol-3-phosphate acyltransferase), MG040 (ABC transporter substrate-binding protein PnrA-like), MG260 mycoplasma lipoprotein, P32 adhesin, P1 adhesin, and a hypothetical protein MG131, each belonging to different protein families (Table 1).

### 3.2. Finalization of Cytotoxic T-Lymphocyte Epitopes

When predicting epitopes by 9-mer length, the MHC-I prediction tool uses the default 2.22 method and the HLA reference set. We analyzed only the epitopes with a percentile rank of 0.5 or greater in the MHC-I binding prediction tool and a %Rank of 1 in NetCTLpan1.1 and NetMHC 4.0. These epitopes were carefully chosen based on their common prediction across all servers. Out of 141 predicted epitopes, five showed high immunogenicity levels from antigenic potential candidate proteins (Table 2) (Appendix A).

### 3.3. Identification of Helper T-Lymphocyte Epitopes

The HLA allele reference set was employed in the projection of epitopes (15-mer in length) using the consensus 2.22 method from the IEDB MHC-II. This tool, along with NetMHCIIpan 3.2, was harnessed to sort the predicted binders by percentile ranks < 0.5. From all of the possible HTL epitopes, a total of 6 were preferred for the final chimeric construct (Table 3).

### 3.4. Probing the Linear BCEs

The ABCPred [35] and BepiPred [37] servers forecasted B-cell epitopes of seven targeted proteins. The analysis generated a total of 850 peptides. The tools found five predicted peptides from four proteins that overlapped, which were taken further for evaluating the candidate vaccine.

### 3.5. Fusion of Final Candidates

The vaccine was designed using high-scoring BCE, CTL, and HTL epitopes in Figure 2. To enhance its immune instigating properties, the GPI anchor protein of *T. cruzi* (Uniport ID: P84883) was merged into the construct using the EAAK linker. Other linking sequences such as AAY, KK, and GPGPG were used to attach epitopes together in a sequential manner (Figure 2).

### 3.6. Physicochemical and Antigenicity Profiling of Chimeric Construct

With ProtParam, a set of physicochemical features for the chimeric vaccines were predicted based on their amino acid sequences. This protein is expected to have a size of 44.7 kDa and a pI of 9.73. Here, 44 and 14 positive and negative charged residues were noted, respectively. An instability index of 29.92 was calculated, indicating a stable molecule for the multiepitope vaccine. An aliphatic index of 96.92 demonstrated the molecule’s high thermal resistance. It is imperative to have a long half-life when heterologous expression is occurring in bacteria or yeast. The half-life of our molecule in mammalian reticulocytes (in vitro) was 30 h, while it was >20 h in yeast and >10 h in *E. coli*, both in vivo. The GRAVY index was found to be 0.161, with the positive value representing the non-hydrophilicity of the designed molecule.

### 3.7. Solubility Analysis

The Protein-Sol server predicted a solubility of 0.494%, which was found out to be greater than the population average for the experimental dataset (PopAvrSol) of 0.45. Therefore, the result indicates the vaccine construct’s solubility as being greater than most *E. coli* proteins (Figure 3). Further confirmation of the solubility was accomplished using the SOLpro server. Compared to the probability of 0.5 on the server, the designed construct had a solubility score of 0.943.

### 3.8. Projection of Vaccine’s 2-D Structure

Using the NPS@ server available from the Prabi server and PSIPRED 4.0, the secondary structure of the protein was determined, where it was found that the designed construct consisted of 59.38% helixes, 32.69% coils, and 6.73% strands (Appendix A). In vaccine constructs, random coils show the occurrence of unfolded regions of proteins that are detectable by antibodies produced in response to infection (Figure 4).

### 3.9. 3D Modeling of the Proposed Construct

The I-TASSER server initiates modeling on the basis of structure templates identified by the PDB library. Rather than generating multiple template alignments, the server takes into account only the most important ones, as determined by the Z-score. Five modelled structures of the antigenic molecule were projected using the ten best templates (0.56 to 5.67). A confidence score value was ascribed to each of the models (−1.48, −2.20, −2.85, −3.14, and −4.38). Typically, the c-scores are between −5 and 2, with a score below −1.5 representing a correct global topology. The model showing the highest c-score of −1.48 was selected as the 3D structure of the chimeric construct (Figure 5A). The predicted TM score for the selected structure was 0.991 and the RMSD value was 0.784 ± 3.7 Å. In order to analyze a protein structure’s similarity to another, the TM score is used, which can diminish all instabilities related to the RMSD. In general, models with TM scores higher than 0.5 have accurate topologies, whereas models with TM scores lower than 0.17 suggest non-specific similarities. Using RoseTTAFold, Robetta predicted the 3D structure of a candidate vaccine; the structure was selected based on a confidence score of 0.52 (Appendix A).

### 3.10. 3D Model Enhancement and Quality Inspection

Using the Ramachandran plot from SAVES, we first compared the models generated by I-TASSER and Robetta. In the comparison, it was found that the initial model from I-TASSER presented 85.03% of the residues in favored regions, while the model generated by Robetta hit 80%. Therefore, the I-TASSER model was preferred for further refinement processes. To enhance the overall structural quality, the GalaxyWeb server was employed (Figure 5B). Out of the five models generated, model 1 was preferred as a result of multiple factors, viz. the GDT-HA (0.9119), RMSD (0.507), and MolProbity (2.375) scores (Appendix A). The clash score was determined as 10.9, the number of bad rotamers as 0.3, and the Ramachandran score as 93%. Therefore, this structure was selected for subsequent studies. According to the Rama plot analysis with PROCHECK, 93%, 4.5%, and 2.5% of the residues reside in preferred, permitted, and disallowed areas, respectively (Figure 5C). Based on the GalaxyRefine scoring procedure, this score is about the same as 93%. In the refined model, ERRAT and ProSA-web were used to assess the global quality and possible errors that may occur. Analyzing the refined model understudy with ERRAT revealed a quality factor of 74.47% (Appendix A) and a z-score of −6.04 from ProSA-web (Figure 5D).

### 3.11. Probing Non-Linear BCEs

In the ElliPro server, 220 residues were mapped to 5 conformational BCEs, with scores fluctuating from 0.56 to 0.84. These epitopes ranged in size from four to eighty-two residues (Table 4) (Figure 6).

### 3.12. Docking Studies of the Modeled Vaccine with Toll-like Receptors

In docking with TLRs, ClusPro created a total of 30 complexes each time with different numbers of cluster members and categorized them according to their weighted scores. For TLR 1, the vaccine construct had the lowest energy cluster of −883.4 (Figure 7A). To further explore the docking analysis, we used HDOCK and Patchdock. HDOCK predicted the binding energies of the vaccine construct with TLR 1 to be −266 (Figure 7B). The PDBsum was used to identify possible residues within MESV that could make stable structural bonds with TLRs (Appendix A). In the designed construct, 13 hydrogen bonds were observed with TLR 1 potential residues within 3 Å (Figure 7C). With the TLR 2- vaccine, the largest cluster had 91 members and the lowest binding energy of −1031.90 (Figure 8A), while HDOCK predicted the binding energies of the TLR 2 vaccine construct to be −270 (Figure 8B). Here, 6 hydrogen bonds were found to be present between TLR 2 and the designed construct (Figure 8C). ClusPro found the lowest binding energy for the TLR 6 vaccine complex to be −880.4 (Figure 9A), whereas HDOCK predicted that the binding energy for the vaccine construct with TLR 6 was −229 (Figure 9B). The interacting residues showed 12 hydrogen bonds (Figure 9C). As determined by FireDock, the refinement of the PatchDock results also generated low global energy values (Table 5).

### 3.13. Binding Energy Analysis

For biological complexes, it is necessary to analyze the binding energies between docked complexes. Based on ΔG, i.e., the binding free energy, the prospect of incidence of contacts at definite circumstances in the cell can be computed. Therefore, using the PRODIGY web server, the binding affinity of the docked complexes was investigated. The input consisted of docked complexes as well as their interactor chains. The temperature was set at 25 °C. For the vaccine-TLR1, vaccine-TLR2, and vaccine-TLR6 complexes, the ΔG values were found to be −10.1 kcal mol^−1^, −11.9 kcal mol^−1^, and −10.9 kcal mol^−1^, respectively. Based on the negative Gibbs free energy values, the results indicated energetically feasible docking. Table 1 provides the complex dissociation constants, the number of interfacial contacts per property, and the non-interacting surfaces per property.

### 3.14. MD Simulations

Molecular dynamic studies are imperative for assessments of protein compactness under a variety of thermobaric conditions. As a result of the GROMACS steepest descent algorithm, the minimum energy for the vaccine construct was obtained to test the overall stability. The energy minimization of a protein is reached when its force is 1000 kJ/mol. A total of 2610 steps was performed for the vaccine construct, where the force reached < 1000 kJ/mol. With a drift of—5.98 × 10^5^ kJ/mol and an average potential energy of—4.35 × 10^5^ kJ/mol, the total potential energy of the system was—4.53 × 10^5^ kJ/mol. A temperature average of 299.8 K was observed after 50,000 steps of NVT with a drift of 1.5 K (Figure 10A). Based on the computed density of the system of 1.004.73 kg/m^3^, the total drift was 0.409284 kg/m^3^. According to Figure 10B, the pressure measured by the system was 1.3 bar with a drift of 2.7 bar. An analysis of the trajectory of the vaccine candidate following a simulation of 50 ns was conducted to test its stability and flexibility. There are only insignificant variations according to RMSD backbone, indicating that the protein is fairly stable (Figure 10C). Based on the radius of gyration, it is apparent that the protein is compact around its axis (Figure 10D). The RMSF plots indicate a high degree of flexibility in vaccine development. This is evident from the high peak values in the plot (Figure 10E).

### 3.15. Immune Simulations

The immune responses in silico generally mirrored the actual results, with the secondary and tertiary responses exceeding the primary responses. As the antibody titers of IgM, IgM + IgG IgG1 + IgG2, and IgG1 increased, the antigen concentrations decreased (Figure 11A). A significant increase in B-cell activity was observed, along with a notable rise in memory B-cells (Figure 11B,C). The Th and Tc cells showed a similar behavior (Figure 11D–F). There was a marked increase in macrophage activity among the innate immune cells (Figure 11G). The IFN-γ and IL-2 levels were also very prominent (Figure 11H). These results suggested that the designed ensemble may induce long-running humoral and cell-mediated immune responses. Finally, the construct was co-adjoined to a PET28(a) vector backbone to test the validity of the in silico cloning (Figure 12).

## 4. Discussions

The skyrocketing antibiotic resistance rates have brought the world dangerously close to the verge of global health crises and are expected to worsen, causing 10 million deaths annually by 2050 [70]. As a result, worldwide efforts have been directed to the development of non-antibiotic therapies, largely focusing on vaccines as effective alternatives against MDR bacteria [71]. While traditional vaccine design methods are time-consuming and expensive, they also require a live pathogen and painstaking immunological, microbiological, and biochemical methods to classify antigenic elements. This poses a safety issue as a consequence [72]. Genomic and proteomic advances have made it possible for computational tools to be used to design the safe and effective vaccines of the future [73]. The application of immunoinformatic approaches to develop vaccines against serotype B meningococcal disease was first witnessed in 2013 with the authorization of the Bexsero and Trumenba vaccines [74]. Several subunit vaccines have been developed for pathogens that are both infectious and difficult to treat using these methods [75,76,77]. Further, subunit-epitope-based vaccinations are potentially useful in preventing microbes from turning pathogenic. We are now able to envisage an accurate vaccine candidate based on the proteome of the pathogen. Generally, this approach leads to efficient, stable, comparatively inexpensive, and benign end products [78].

Our study represents a multivalent subunit vaccine design approach based on a proteome-wide immunoinformatic analysis of immunodominant epitopes against *M. genitalium*. In this study, we screened all 483 proteins of *M. genitalium* using different web-based servers to locate the most immunodominant mycoplasma antigens. Parallel studies have also been performed [79,80]. This is the first study of its kind to explore a possible vaccine candidate against *M. genitalium*. Ali et al. anticipated a possible vaccine structure but did not demonstrate its robustness under realistic life-like conditions [81]. The development of computational vaccines utilizing newly identified virulent antigenic proteins is a promising idea [82]. Considering their role in pathogen adhesion to host cells and in virulence, membrane and extracellular proteins were taken into consideration when choosing epitopes and designing effective MEVs. Human homologs were discarded to avoid autoimmune responses, as well as paralogous, cytoplasmic, and non-essential proteins, which are the least significant. Additionally, the downstream analysis was limited to only highly antigenic proteins. After a detailed physicochemical analysis of antigenic proteins, epitopes for T-cells and B-cells were forecasted. The linkers and adjuvant helped in designing the MEV. The amine terminus of the MEV was adjuvanted via the EAAK linker, whereas the epitopes were adjoined via AAY, KK, and GPGPG linkers. The adjuvant was added to intensify the immunogenic potential of the vaccine [83]. In order to maintain the functions of each epitope when imported into the human body the linkers were incorporated [84].

The immunogenic, non-toxic, and non-allergic properties of MEV highlighted its potential to turn on an effective immune response as an epitope-based vaccine. A physicochemical analysis of the construct revealed it was non-hydrophilic and within the target size range of 110 kDa. By presenting the maximum percentages of residues within the favorable zone, the structure analysis emphasized their structural integrity. A higher level of response by both arms of the immune system was apparent throughout the computational immune response analysis, which corresponded to typical immune responses. Several months of memory formation was observed in B cells. The potential energy of the ensemble was minimized to achieve conformational stability. By replacing some of the protein atoms, energy minimization can repair the unnecessary geometry of the structure, resulting in a more stable structure that is stereochemically compatible. Based on the MDS analysis and subsequent molecular docking studies, the MEV proved to be very stable and capable of tightly binding with the TLR receptors. Through codon optimization, the expression of the MEV construct within *E. coli* K12 was further enhanced. Based on the results of our research, MEV appears to be a promising candidate for further in vitro and in vivo analyses in order to develop a potentially effective vaccine against *M. genitalium*. Through the production of host-defensive T- and B-cells in the mucosa and surrounding system, it would boost immune responses in the mucosal membrane, preventing pathogens from entering the host. Thus, the MEV design allows for the activation of an effective immune response using a minimal, well-defined antigen. In vitro testing can be performed on the designed construct with the same design or with minor modifications to enhance its performance.

## 5. Conclusions

The rapidly increasing life-threatening infections caused by *Mycoplasma genitalium* are becoming onerous to combat. For patients infected with this infection, there is no proper medical prevention, such as a vaccine. With the help of in silico methods, it is possible to cost-effectively and quickly design an effective vaccine. In this study, a multiepitope vaccine consisting of conserved CTL, HTL, and B-cell epitopes, which can activate strong immune responses, was developed against *M. genitalium*. The designed multiepitope vaccine showed high immunogenic potential. By using a molecular dynamics simulation, the vaccine’s stability was assured, while docking studies confirmed its stability with immune receptors. Studying the in silico expression of the vaccine in bacteria confirmed its expression, while the vaccine’s ability to trigger an immune response was validated by simulation studies. Since the vaccine candidates identified are highly conserved among pathogenic mycoplasmas and their potential as vaccine candidates has not been adequately assessed, they may deserve further examination as potential vaccine candidates. Moreover, to validate their effectiveness as vaccine candidates against *M. genitalium*-induced STIs, further experimental studies are needed, including with in vitro tests and animal models.

## Figures and Tables

**Figure 1 vaccines-10-01720-f001:**
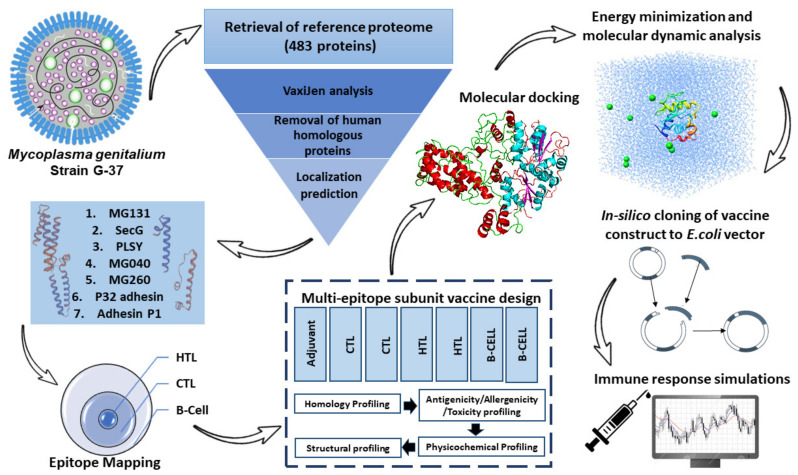
Graphical representation of methods used to project the multiepitope protein subunit and its subsequent characterization.

**Figure 2 vaccines-10-01720-f002:**
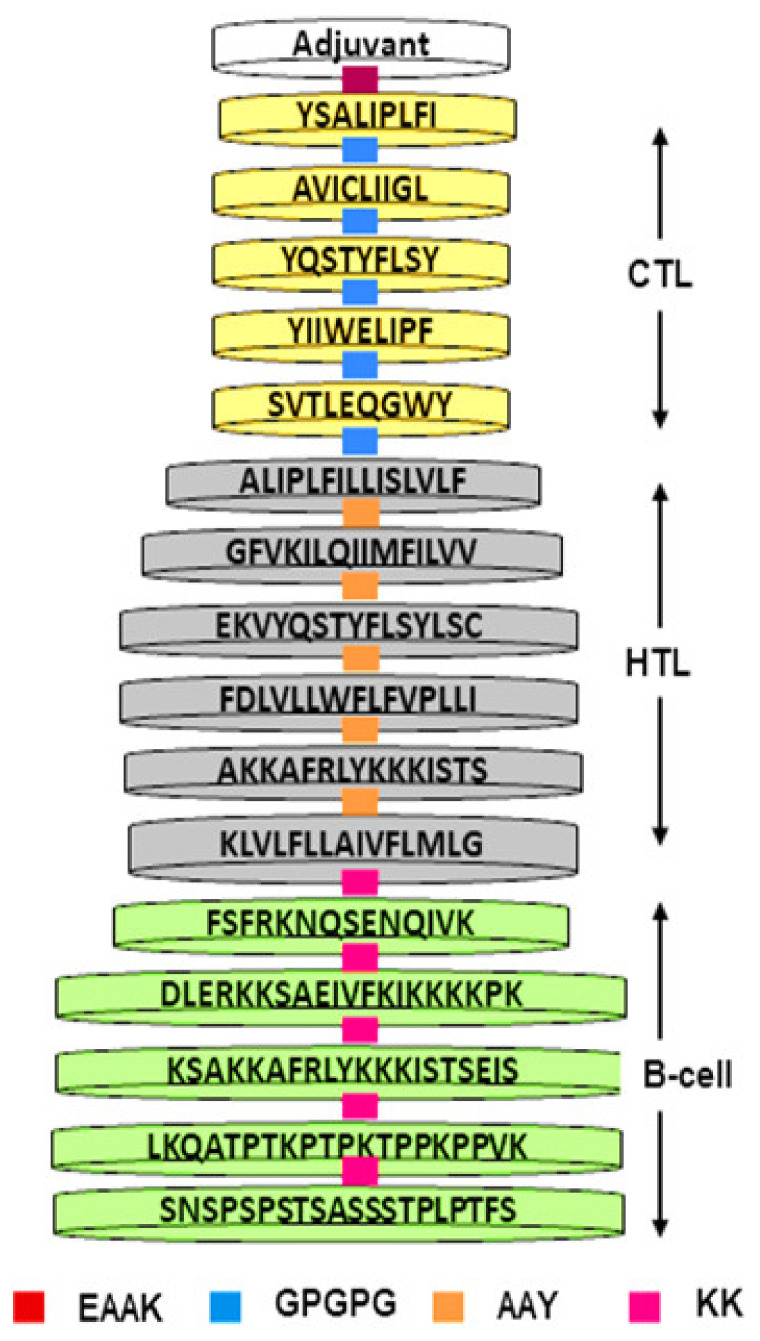
Scheme of the designed chimeric consisting of fused T cells joined via GPGPG and AAY and B cell epitopes connected via KK linkers.

**Figure 3 vaccines-10-01720-f003:**
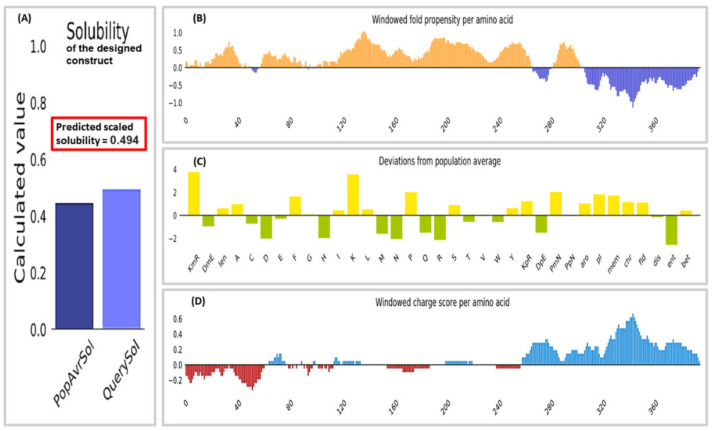
Solubility analysis of the multiepitope construct conducted using the protein-sol server. (**A**) Solubility plot of the designed construct alongside the population average for the investigational dataset. The solubility of the designed construct was calculated to be 0.494. (**B**) Windowed fold propensity. (**C**) Plot shows deviations from population averages for 35 features. (**D**) Windowed net charge.

**Figure 4 vaccines-10-01720-f004:**
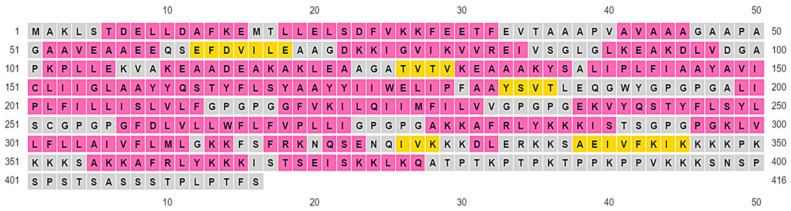
Color-coded annotation grid representing the secondary elements of the antigenic chimera as produced by PSIPRED 4.0. Yellow: Strands, Pink: Helix, Gray: Coils.

**Figure 5 vaccines-10-01720-f005:**
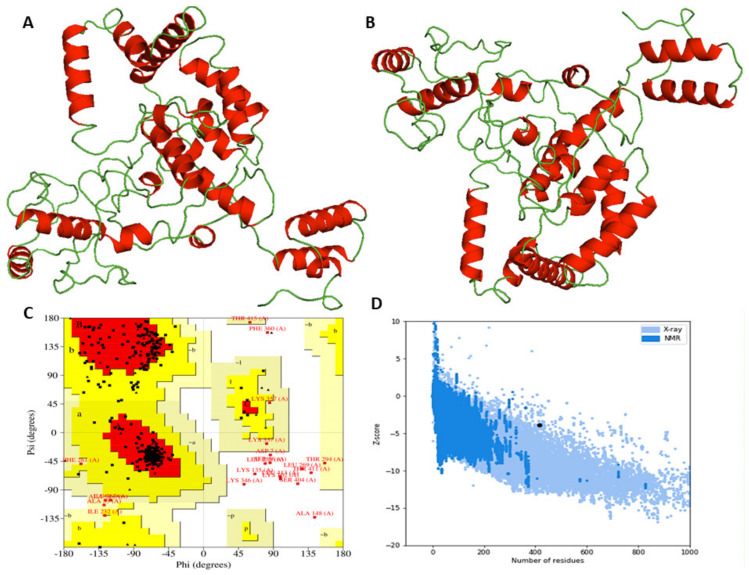
Modeling, fine-tuning, and quality inspection of 3D models: (**A**) chimeric protein tertiary model created via I-TASSER; (**B**) refined model using Galaxy-web; (**C**) Ramachandran plot of vaccine construct; (**D**) Z-score (−6.04) as projected by ProSA-Web.

**Figure 6 vaccines-10-01720-f006:**
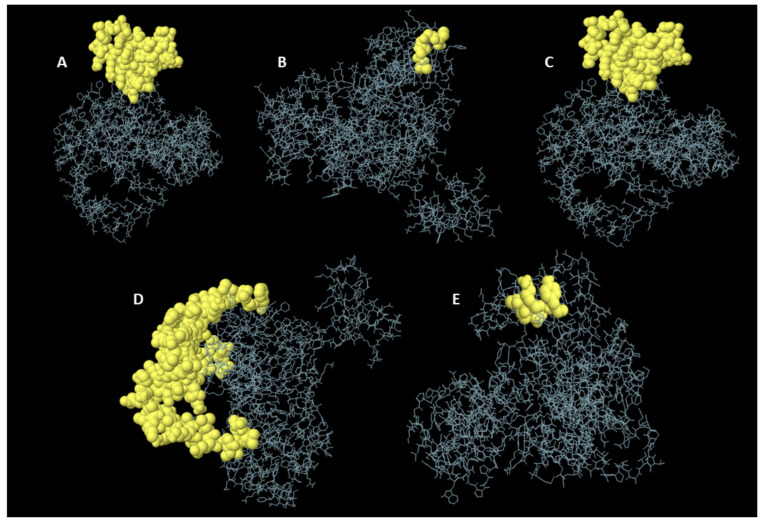
Discontinuous BCEs as projected via the Ellipro webserver: (**A**–**E**) illustrations of the non-linear BCEs present in the designed construct from various angles, with yellow showing the epitopes and grey showing the bulk of the protein.

**Figure 7 vaccines-10-01720-f007:**
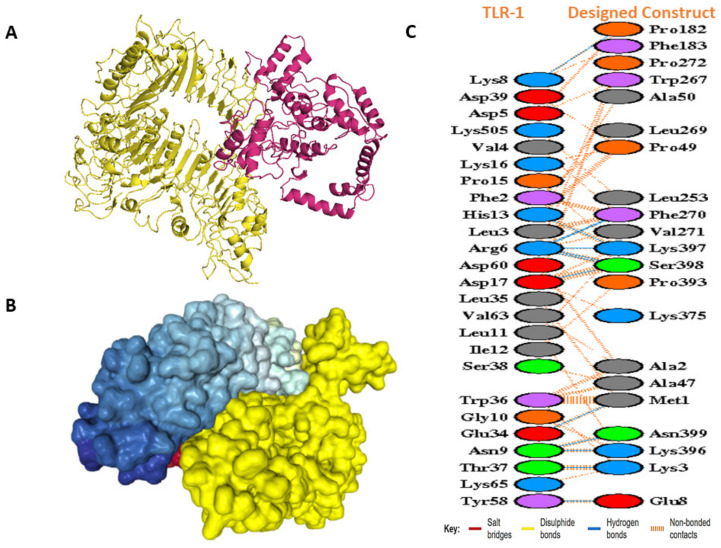
Docking studies of the designed construct with TLR 1. (**A**) Docked complex as generated by the ClusPro webserver. The yellow ribbon structure reflects the TLR1 molecule, while the warm pink represents the vaccine molecule. (**B**) Vaccine’s docked complex generated by HDOCK (yellow represents vaccine). (**C**) Residue interactions between TLR 1 and the vaccine construct. Hydrogen bonds are specified with blue color lines.

**Figure 8 vaccines-10-01720-f008:**
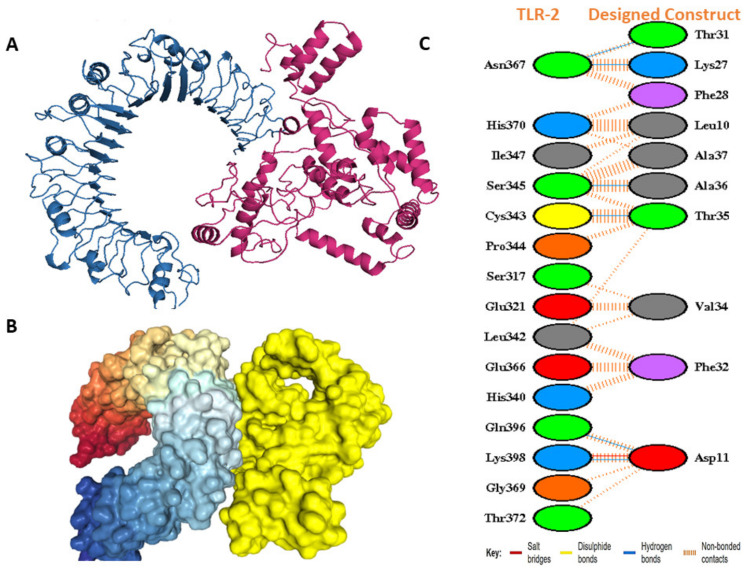
Docking studies of the designed molecule with TLR 2. (**A**) ClusPro-generated docked complex. The blue ribbons symbolize the TLR 2 molecule, while the warm pink corresponds to the vaccine molecule. (**B**) HDOCK docked complex (yellow represents vaccine). (**C**) Residue interactions between TLR 2 and vaccine construct. Hydrogen bonds are revealed with blue lines.

**Figure 9 vaccines-10-01720-f009:**
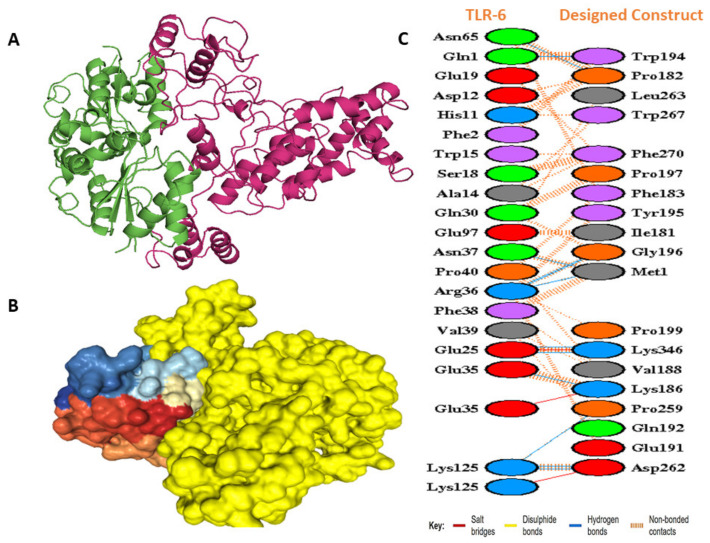
Molecular docking studies of the designed molecule with TLR 6. (**A**) Docked complex as predicted by ClusPro. The green ribbons depict the TLR 6 molecule, while the warm pink corresponds to the vaccine molecule. (**B**) HDOCK docked complex (yellow represents vaccine). (**C**) Residue interactions between TLR 6 and vaccine construct. Hydrogen bonds are indicated with blue lines.

**Figure 10 vaccines-10-01720-f010:**
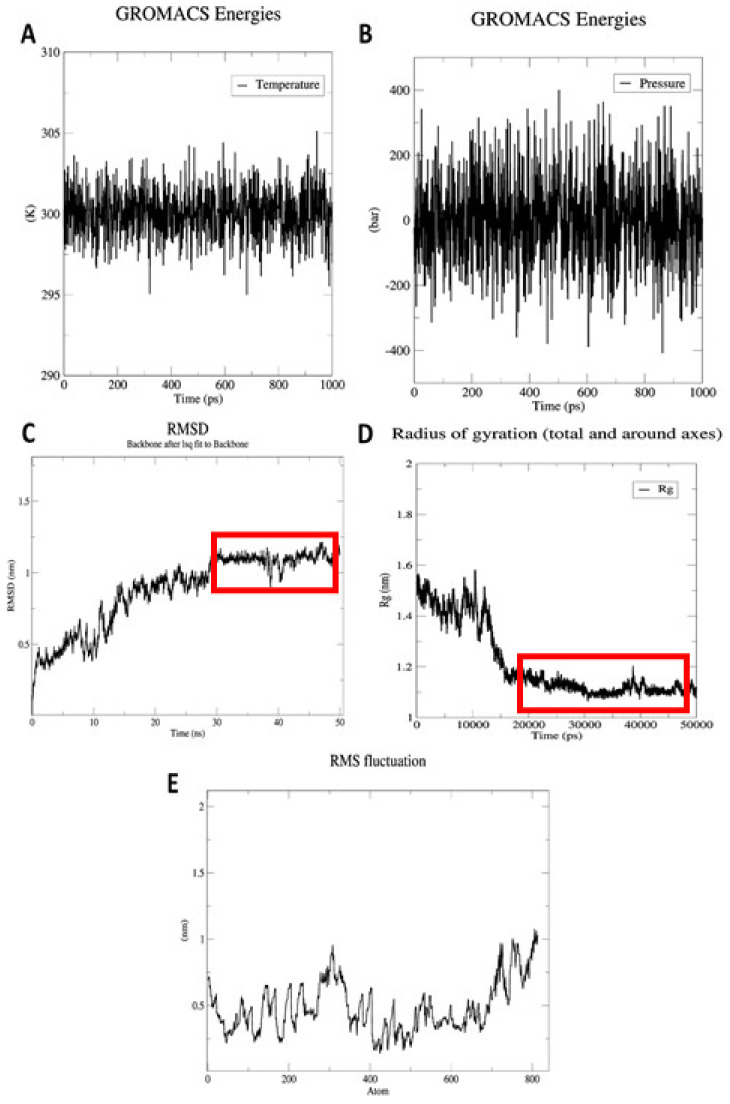
Graphs engendered throughout the MD simulation stages. (**A**) Temperature plot. System successfully achieves temperature of 300 K and showed least fluctuations afterwards. (**B**) Pressure plot. System maintains the pressure at 1.3 bar throughout 100 ps. (**C**) RoG plot. RoG is fairly stable after 20 ns, showing the vaccine in its compacted form during the simulation run. (**D**) RMSD plot. RMSD of the protein backbone reaches ~1.2 nm after 30 ns and is maintained generally afterwards, which represents the minimum structural deviations of a vaccine construct. (**E**) RMSF plot. Side chain’s RMSF plot displays peaks depicting high flexibility in certain areas of the designed construct.

**Figure 11 vaccines-10-01720-f011:**
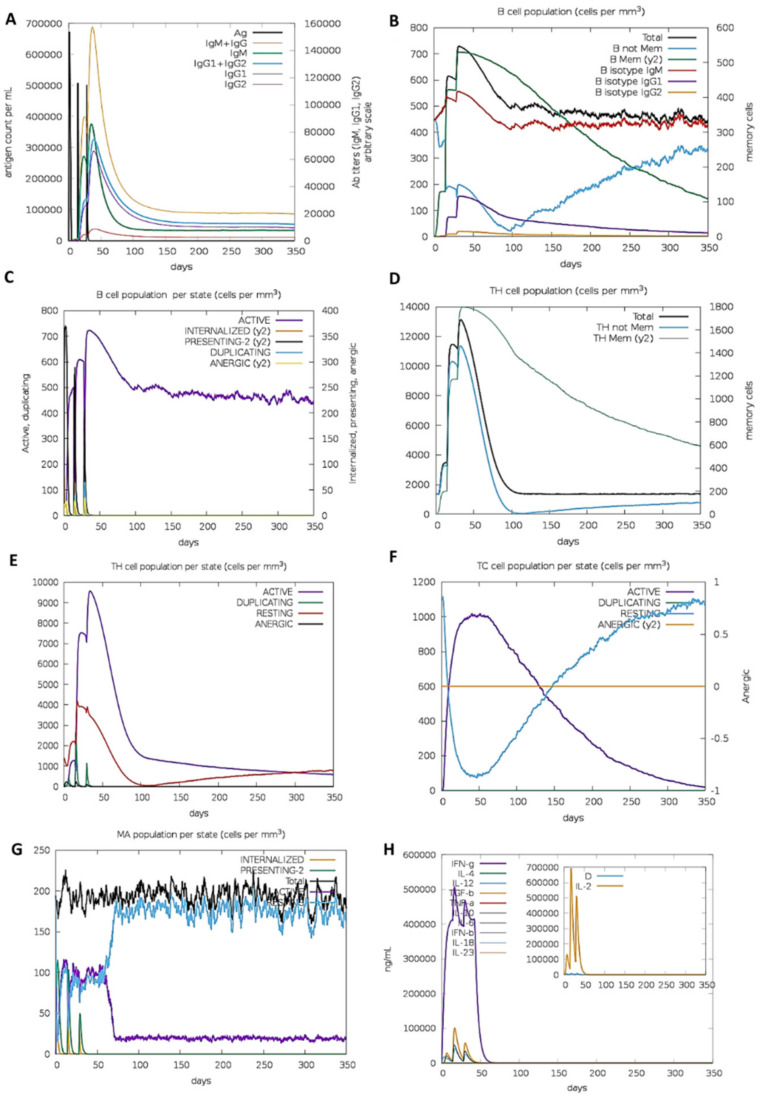
Simulated immune responses generated by the designed construct: (**A**) antibody titers following antigen injections; (**B**,**C**) memory and non-memory B-cell populations, respectively; (**D**,**E**) TH cell population and population per state, respectively; (**F**) TC cell population per state; (**G**) MA population per state; (**H**) Simpson index “D” and cytokine level.

**Figure 12 vaccines-10-01720-f012:**
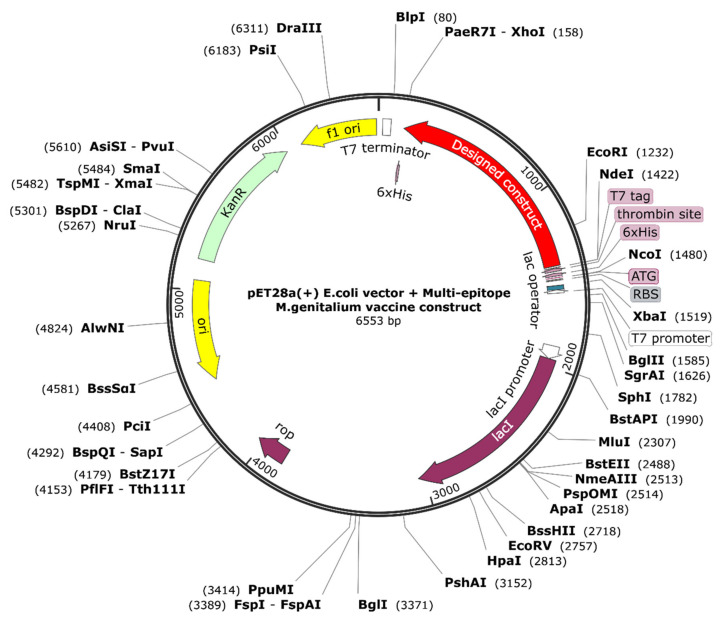
In-silico-cloned vaccine construct for *E. coli* expression system, colored with a red arrow, shown in a black vector backbone of pET28 (a).

**Table 1 vaccines-10-01720-t001:** Most immunogenic proteins selected as potential vaccine candidates in *M. genitalium* proteome.

	Uniprot Entry Identifier	Protein	Antigenicity (VaxiJen)	Non-Allergen	Subcellular Localization
1	P47377	MG131 (Hypothetical protein)	1.16	Yes	Cytoplasmic membrane
2	P58061	SECG (Probable protein-export membrane protein SecG)	0.93	Yes	Cytoplasmic membrane
3	P47489	PLSY (Glycerol-3-phosphate acyltransferase)	0.92	Yes	Cytoplasmic membrane
4	P47286	MG040 (ABC transporter substrate binding protein pnrA-like)	0.76	Yes	Cytoplasmic membrane
5	P47502	MG260 (Mycoplasma lipoprotein)	0.76	Yes	Cytoplasmic Membrane
6	Q49417	P32 adhesin	0.69	Yes	Cytoplasmic Membrane
7	P20796	Adhesin P1	0.52	Yes	Cytoplasmic Membrane

**Table 2 vaccines-10-01720-t002:** Conserved and most immunogenic CTL epitopes.

Protein ID	CTL Epitopes (9-mer)	MHC Class(I) Supertypes	Binding Affinity	VaxiJen Score	Non-Allergen	Toxicity	Conservation
P47377	YSALIPLFI	A1, A24	0.24	1.8319	Yes	No	100%
P58061	AVICLIIGL	A1, A2	0.57	0.82	Yes	No	100%
P47489	YQSTYFLSY	A3, A24	0.17	1.22	Yes	No	100%
P47286	YIIWELIPF	A1, A24	0.11	2.1807	Yes	No	100%
P47502	SVTLEQGWY	A2, A3, A24	0.46	0.82	Yes	No	100%

**Table 3 vaccines-10-01720-t003:** Conserved and most immunogenic HTL epitopes.

Protein ID	HTL Epitopes (15-mer)	Percentile Rank	IC50	VaxiJen Score	Non-Allergen	Toxicity	Conser- Vation
P47377	ALIPLFILLISLVLF	0.01	0.21	2.11	Yes	No	100%
P58061	GFVKILQIIMFILVV	0.02	0.32	0.82	Yes	No	100%
P47489	EKVYQSTYFLSYLSC	0.04	0.41	0.59	Yes	No	100%
P47286	FDLVLLWFLFVPLLI	0.01	0.69	3.53	Yes	No	100%
P47502	AKKAFRLYKKKISTS	0.03	0.81	0.51	Yes	No	100%
P20796	KLVLFLLAIVFLMLG	0.15	0.95	1.29	Yes	No	100%

**Table 4 vaccines-10-01720-t004:** Conformational BCEs projected using ElliPro. A total of 220 residues spanned across five discontinuous BCE regions of the refined model.

No.	Residues	Total Residues	Score
1	A:M1, A:A2, A:K3, A:L4, A:S5, A:T6, A:D7, A:E8, A:L9, A:L10, A:D11, A:A12, A:F13, A:K14, A:E15, A:M16, A:T17, A:L18, A:L19, A:E20, A:L21, A:S22, A:D23, A:F24, A:V25, A:K26, A:K27, A:F28, A:E29, A:E30, A:T31, A:F32, A:E33, A:V34, A:T35, A:A36, A:A37, A:A38, A:P39, A:V40, A:A41, A:V42, A:A43, A:A44, A:A45, A:G46, A:A47, A:A48, A:P49, A:P381	50	0.841
2	A:E68, A:A69, A:G71, A:D72, A:K73, A:I75, A:G76, A:V77, A:I78, A:K79, A:V80, A:V81, A:R82, A:E83, A:I84, A:V85, A:S86, A:G87, A:L88, A:G89, A:L90, A:K91, A:E92, A:A93, A:K94, A:D95, A:L96, A:V97, A:D98, A:G99, A:A100, A:P101, A:K102, A:P103, A:L104, A:L105, A:E106, A:K107, A:V108, A:A109, A:K110, A:E111, A:A112, A:A113, A:D114, A:E115, A:A116, A:K117, A:A118, A:L120, A:E121, A:A122, A:G124, A:A125, A:T126, A:T128, A:K290, A:K291, A:I292, A:S293, A:T294, A:S295, A:G296, A:P297, A:G298, A:P299, A:G300, A:K301, A:L302, A:V303, A:L304, A:F305, A:L306, A:A308, A:I309, A:V310, A:L312, A:M313, A:L314, A:G315, A:F316, A:S317	82	0.699
3	A:W178, A:E179, A:L180, A:I181, A:P182, A:F183, A:A184, A:A185, A:K186, A:S187, A:V188, A:T189, A:L190, A:E191, A:Q192, A:G193, A:W194, A:Y195, A:G196, A:P197, A:G198, A:P199, A:G200, A:A201, A:I207, A:S211, A:L212, A:V213, A:L214, A:F215, A:G216, A:P217, A:G218, A:P219, A:F222, A:V223, A:K224, A:I225, A:L226, A:Q227, A:I228, A:I229, A:M230, A:F231, A:I232, A:L233, A:V234, A:V235, A:G236, A:P237, A:G238, A:P239, A:G240, A:E241, A:K242, A:V243, A:Y244, A:Q245, A:S246, A:T247, A:Y248, A:F249, A:L250, A:S251, A:Y252, A:L253, A:S254, A:C255, A:G256, A:P257, A:G258, A:P259, A:G260, A:F261, A:D262, A:L263, A:L266, A:W267	78	0.687
4	A:D64, A:V65, A:I66, A:E334	4	0.576
5	A:K130, A:A132, A:A133, A:A134, A:K135, A:Y139	6	0.56

**Table 5 vaccines-10-01720-t005:** Docking result produced by Patchdock and refined by Firedock.

**Vaccine Construct**	**TLRs**	**Solution**	**^a^ GBE**	**^b^ aVdW**	**^c^ HBE**	**^d^ ACE**	**^e^ Score**	**^f^ Area**
	TLR 1	1	−37.53	−40.86	−2.37	4.44	20,226	3124.50
	TLR 2	7	−12.41	−52.34	−3.94	−0.55	17,078	2019.00
	TLR 6	4	−7.79	−24.19	−5.19	8.37	15,410	2584.60

^a^ Global binding energy. ^b^ Attractive Van der Waals forces. ^c^ Hydrogen bond energy. ^d^ Atomic contact energy. ^e^ Geometric shape complementarity score. ^f^ Estimated interface area of the complex.

## Data Availability

Not Applicable.

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
