# Peer review of "Vaccinomics-Aided Development of a Next-Generation Chimeric Vaccine against an Emerging Threat: Mycoplasma genitalium"

_vaccines, 2022, doi:10.3390/vaccines10101720_

Round 1
Reviewer 1 Report
Kashaf Khalid et al utilized reverse vaccinology to develop a chimeric vaccine against Mycoplasma genitalium by several bioinformatic tools identifying vaccine targets of this pathogen. but their potential as vaccine candidates has not been adequately assessed in-vitro tests and animal models.More specific comments are below.
Comments and Suggestions for Authors
1. Line 120 The reference proteome of M.genitalium (Strain G-25 37) was used for identifying vaccine targets of this pathogen, but are there different serotypes of M. granitalium? Is there cross protection between different serum strains? Or are the genomes or major antigenic proteins conserved among strains of different serotypes?These information is not clear, please add them.
2. Since Mycoplasma genitalis mainly affects the urethra or reproductive tract, mucosal immunity should be considered in vaccine design.
3. Table 1 seven antigenic proteins were screened as potential vaccine candidates in m.genitalium,but which of them have been confirmed as candidate antigens and which of them are newly identified. please provide additional background information and discuss them.
4. Indeed, the arrangement between epitopes may affect the conformation and antigenicity of epitopes,Figure 2 shows only one final candidate of epitopes arrangement,From which permutations was the final combination selected? Please provide details of the screening process and analysis data.but the final vaccine candidates has not been adequately assessed in-vitro tests and animal models.Multiple candidate combinations should be provided for further screening in subsequent animal studies.
Author Response
We are thankful to the editor for his considerations and positive response. We have replied to the comments of worthy reviewer (comments are differentiated in blue color), made necessary changes in the manuscript where required, and summarized them all below. The yellow highlighted italics part in the document represents the modification that we have made in our manuscript.
Reviewer# 1
The authors are grateful for the time and efforts taken by the worthy reviewer in reviewing the manuscript and giving constructive comments. We have revised the manuscript following the suggestions and observations made by the reviewer. We hope that the revised manuscript will meet the publication requirements of the journal.
Comments and Suggestions for Authors
- Line 120 The reference proteome of M.genitalium (Strain G-25 37) was used for identifying vaccine targets of this pathogen, but are there different serotypes of M. granitalium? Is there cross protection between different serum strains? Or are the genomes or major antigenic proteins conserved among strains of different serotypes? These information is not clear, please add them.
Response: We are thankful to the reviewer for keen analysis and this comment. As a matter of fact, M.genitalium does not have different serovars. There are different strains, out of which G-37 is the most common and well-studied, therefore termed as the reference strain. The designed vaccine will provide cross-protection across similar strains.
There are different mycoplasmas in humans but those are not as much clinically important as M.genitalium(Fookes et al., 2017).
To clarify the point, following changes have been to the file.
The reference proteome of M.genitalium (G-37 / strain ATCC 335300/ NCTC 10195) was retrieved from the UniProt[29]. The reference strain was selected because it is well conserved, more common, and clinically significant in comparison to other M.genitalium and mycoplasmas. A file containing the FASTA sequences of each of the proteins was then uploaded to the VaxiJen server that employs an alignment-independent algorithm based on primary amino acid characteristics[30].
Reference
Fookes, M.C. et al. (2017) ‘Mycoplasma genitalium: whole genome sequence analysis, recombination and population structure’, BMC Genomics, 18(1). doi:10.1186/S12864-017-4399-6.
- Since Mycoplasma genitalis mainly affects the urethra or reproductive tract, mucosal immunity should be considered in vaccine design.
Response: We sincerely acknowledge the reviewer’s comment. The designed construct has shown to triggering the host immune cells (T and B cells) and therefore it would successfully trigger the immune cells generation in the mucosal membranes too. To emphasize this point, we have added the following lines in the discussion section.
Based on results of our research, MEV appears to be a promising candidate for further in vitro and in vivo analysis in order to develop a potentially effective vaccine against M.genitalium. Through the production of host-defensive T- and B-cells in the mucosa and surrounding system, it would boost immune responses in the mucosal membrane, preventing pathogens from entering the host. Thus, the MEV design allows for the activation of an effective immune response using a minimal, well-defined antigen.
- Table 1 seven antigenic proteins were screened as potential vaccine candidates in m.genitalium, but which of them have been confirmed as candidate antigens and which of them are newly identified. please provide additional background information and discuss them.
Response: We highly thank the reviewer for pointing this out. To clarify this question, we have added following lines to the results and discussion section
Results
As a result, new antigenic proteins were found that have not yet been analyzed. These consist of SecG (protein-export membrane protein), PLSY (Glycerol-3-phosphate acyltransferase), MG040 (ABC transporter substrate binding protein pnrA-like), MG260 Mycoplasma lipoprotein, P32 adhesin, P1 adhesin and a hypothetical protein MG131, each belonging to different protein families (Table 1).
Discussion
The development of computational vaccines utilizing newly identified virulent antigenic proteins is a promising idea (Naz et al., 2015)
- Indeed, the arrangement between epitopes may affect the conformation and antigenicity of epitopes. Figure 2 shows only one final candidate of epitopes arrangement. From which permutations was the final combination selected? Please provide details of the screening process and analysis data but the final vaccine candidates has not been adequately assessed in-vitro tests and animal models. Multiple candidate combinations should be provided for further screening in subsequent animal studies.
Response: We agree with the reviewers comment. However, the finality of the construct was determined after antigenicity and allergenic testing. The structure showing the highest antigenic value and non-allergenic behavior was selected for vaccine formulation. We have added the following lines to discussion section.
In-vitro testing can be performed on the designed construct with the same design or with minor modifications to enhance its performance.
Reviewer 2 Report
The present study entitled “Vaccinomics-aided development of next generation chimeric vaccine against emerging threat: Mycoplasma genitalium” shows the in silico construction of protein vaccine against Mycoplasma genitalium. The study also shows the in silico representations of effective antigenicity and immunogenic efficiencies of the vaccine. The article is well written with all plausible representations. The following comments may be considered:
Minor comments:
1. In the Abstract, Line no. 26-27, page no. 1, “Using proteins that are non-toxic, non-allergic and non-homologous to the human proteome, a multi-epitope vaccine was developed” Please reconstruct the sentence in simple way. Please replace “proteome” with “proteins”.
2. Line no. 58-59, page no. 2, “M.genitalium is a slow grower, …………. facultative anaerobe”. “slow grower” should be replaced by “slow-growing”.
3. Line no. 99-113, page no. 3, Please check the paragraphs and remove.
4. Please discuss the safety concerns of the vaccine with supportive data.
5. Line no. 607-608, page no. 23, Reference No: 24, Name of Journal is missing. Please include.
Author Response
We are thankful to the editor for his considerations and positive response. We have replied to the comments of worthy reviewer (comments are differentiated in blue color), made necessary changes in the manuscript where required, and summarized them all below. The yellow highlighted italics part in the document represents the modification that we have made in our manuscript.
Reviewer# 2
The present study entitled “Vaccinomics-aided development of next generation chimeric vaccine against emerging threat: Mycoplasma genitalium” shows the in silico construction of protein vaccine against Mycoplasma genitalium. The study also shows the in silico representations of effective antigenicity and immunogenic efficiencies of the vaccine. The article is well written with all plausible representations. The following comments may be considered:
Minor comments:
- In the Abstract, Line no. 26-27, page no. 1, “Using proteins that are non-toxic, non-allergic and non-homologous to the human proteome, a multi-epitope vaccine was developed” Please reconstruct the sentence in simple way. Please replace “proteome” with “proteins”.
Response: Dear reviewer, thank you for pointing this out. We have rephrased the sentence into a simpler form as follows:
A multi-epitope vaccine was developed using proteins that are not toxic, non-allergic, and non-homologous to human proteins.
- Line no. 58-59, page no. 2, “M.genitalium is a slow grower, …………. facultative anaerobe”. “slow grower” should be replaced by “slow-growing”.
Response: We are grateful to the reviewer for this comment and therefore the comment has been addressed.
- Line no. 99-113, page no. 3, Please check the paragraphs and remove.
Response: We apologize for our negligence. The paragraphs have been removed.
- Please discuss the safety concerns of the vaccine with supportive data.
Response: The immunoinformatic profile of the vaccine shows that it is safe to use. However, experimental evaluations can help in further testing the safety profile of the vaccine. Therefore, in order to highlight this point, we have added following statement in the discussion section.
A limitation of this study is validating the final vaccine construct experimentally before it can be used in clinical practice.
- Line no. 607-608, page no. 23, Reference No: 24, Name of Journal is missing. Please include.
Response:
We are sincerely grateful to the reviewer for the comment.The journal name has been added as follows:
- Taylor-Robinson, D. Diagnosis and Antimicrobial Treatment of Mycoplasma Genitalium Infection: Sobering Thoughts. Expert Rev. Anti. Infect. Ther. 2014, 12, 715–722.
Round 2
Reviewer 1 Report
-
I agree with the revision of the manuscript.